Resistance characterization and transcriptomic analysis of imipenem-induced drug resistance in Escherichia coli

Tong Chunyu
Liang Yimin
Liu Qi
Yu Honghao
Feng Wenzhi
Song Bocui songbocui66@163.com
College of Life Science and Technology, Heilongjiang Bayi Agricultural University , Daqing , Heilongjiang , China
Zhang Xin
Electronic publication date: 2024 Nov 29
Publication date: 2024
Volume: 12
Electronic Location ID: e18572
Received 2024 Jun 21; Accepted 2024 Nov 1
Copyright: ©2024 Tong et al.
Copyright year: 2024
Copyright holder: Tong et al.
License: This is an open access article distributed under the terms of the Creative Commons Attribution License, which permits unrestricted use, distribution, reproduction and adaptation in any medium and for any purpose provided that it is properly attributed. For attribution, the original author(s), title, publication source (PeerJ) and either DOI or URL of the article must be cited.
License URL: https://creativecommons.org/licenses/by/4.0/

Keywords: Escherichia coli, Imipenem, Transcriptomics, Biofilm, Efflux pumps

Funding: Key Research and Development Project of Heilongjiang Province of China GZ20210101 Cultivation Project of Heilongjiang Bayi Agricultural University XDB-2016-22 Postdoctoral Scientific Research Start-up Fund of Heilongjiang LBH-Q21158 This research was funded by the Key Research and Development Project of Heilongjiang Province of China, grant number GZ20210101; the Cultivation Project of Heilongjiang Bayi Agricultural University, grant number XDB-2016-22; the Postdoctoral Scientific Research Start-up Fund of Heilongjiang, grant number LBH-Q21158. The funders had no role in study design, data collection and analysis, decision to publish, or preparation of the manuscript.

==============================
Background

Bacteria can develop resistance to various antibiotics under selective pressure, leading to multifaceted changes in resistance mechanisms. Transcriptomic sequencing allows for the observation of transcriptional level alterations in cells under antibiotic stress. Understanding the bacterial response to such stress is essential for deciphering their strategy against drug-resistant antibiotics and identifying potential targets for antibiotic development.

Methods

This study using wild-type (WT) Escherichia coli (E. coli) discovered that continuous in vitro induction screening for imipenem-resistant strains resulted in bacteria with enhanced biofilm-forming ability and mutations in antibiotic target sites. Transcriptomic sequencing of the resistant bacteria revealed significant changes in carbon and amino acid metabolism, nutrient assimilation, substance transport, nucleotide metabolism, protein biosynthesis, and cell wall biosynthesis. The up-regulated drug efflux genes were disrupted using gene knockout technology. Drug sensitivity tests indicated that drug efflux has a minimal effect on imipenem resistance.

Results

This suggests a strategy for E. coli drug resistance involving the reduction of unnecessary substance synthesis and metabolism, coupled with an increase in activities that aid in resisting foreign threats.

Introduction

The increasing prevalence of multidrug-resistant (MDR) Gram-negative pathogens, often involving β-lactamase production, is a significant global concern (Barbier, Lipman & Bonten, 2016). Carbapenems, being broad-spectrum β-lactam antimicrobials highly stable against β-lactamases and cephalosporinases, have become crucial antibiotics for treating β-lactamase-resistant bacteria due to the rise in resistance to fluoroquinolone antibiotics (El-Gamal et al., 2017; Ong’uti et al., 2022). Subsequently, other carbapenem antibiotics such as ertapenem, meropenem, and doripenem have been developed and are now widely used in clinical practice (Potter, D’Souza & Dantas, 2016). However, the global emergence and spread of carbapenem-resistant pathogens have intensified the clinical challenges in treating these infections (Onishi et al., 2021; Mancuso et al., 2023).

The construction of resistant strain models through drug induction in vitro is a common method used to predict the emergence and progression of bacterial resistance to drugs in clinical settings, as well as to study resistance mechanisms (Zhou et al., 2019; Geng et al., 2022). Generally, E. coli exhibits the following resistance mechanisms (Christaki, Marcou & Tofarides, 2020): (1) altering membrane permeability to resist drugs; (2) producing modifying enzymes that disrupt the chemical structure of antimicrobial drugs, rendering them ineffective; (3) mutating the target of antimicrobial drugs, changing the sites of action on the bacterial surface and interior or creating new sites, preventing the drug from binding effectively; (4) increasing expression of drug efflux transporters, enhancing drug efflux and leading to drug resistance.

Non-pathogenic E. coli, similar to pathogenic bacteria, exhibit stress responses to sublethal environmental triggers. These responses aid in bacterial survival and potentially enhance virulence (Yang et al., 2020). Investigating changes in gene expression in response to stress is crucial for understanding disease (Nagar et al., 2016; Harms, Maisonneuve & Gerdes, 2016). Valat et al. (2020) explored the impact of a low dose of ciprofloxacin on the E. coli O26:H11 transcriptome through a combination of RNA-seq and RT-qPCR. They found changes in the SOS response, the type III secretion system (T3SS) effector expression, and dynamics regulation. Bie et al. (2023) conducted a detailed comparative transcriptomic analysis of E. coli’s response to nine antibiotics, aiming to understand how this model organism reacts at the transcriptional level. Enrichment analyses on GO, KEGG, and EcoCyc pathways highlighted significant changes in various metabolic pathways, stress response, protein synthesis, and cell communication.

In this study, the carbapenem drug imipenem was utilized to continuously induce antibiotic resistance in E. coli standard strain K-12 MG1655 in vitro, resulting in the development of an imipenem-resistant strain. We conducted biological verification of the aforementioned resistance pathways and performed transcriptomic sequencing of the drug-resistant bacteria. Aimed to elucidate the changes in transcription levels that occurred in the bacteria following the development of drug resistance and to identify the primary mechanisms employed to counteract imipenem exposure.

Materials & Methods

Induction of imipenem-resistant E. coli

This study utilized WT E. coli K-12 MG1655 to induce drug-resistant strains using the 1/2 × MIC (minimum inhibitory concentration) drug induction method. The induction process, outlined in Fig. 1, involved inoculating the E. coli culture into MH medium containing 1/2 × MIC-inducing drug and incubating at 37 °C for 24 h. Subsequently, the induced strain was exposed to a drug concentration gradient ranging from 32 to 0.008 µg/mL at the same temperature for another 24 h. The MIC value of the induced strain was determined by observing its growth status and monitoring the changes in MIC concentration over multiple generations. The drug susceptibility of the induced strain was assessed using the disc diffusion method with imipenem against the WT strain, the drug-resistant strain, and a quality control strain following CLSI 2022 guidelines. Each disc had a concentration of 10 µg, and the diameter of the inhibitory zone was measured to evaluate the efficacy of imipenem.

Figure 1 1/2MIC concentration gradient induces the development of drug-resistant bacteria.

Allow the strain to grow in MH medium containing a sub-inhibitory concentration of the drug. As the minimum inhibitory concentration (MIC) increases, gradually elevate the drug concentration in the medium until the MIC stabilizes and no longer changes.

Cross-resistance testing

To investigate the cross-resistance of IPM-R strains, this study selected 12 antibiotics. Table 1 displays the names and resistance mechanisms of these antibiotics (Houang & Greenwood, 1977). The MIC values of the resistant strain to antibiotics were determined using the broth microdilution method, with the WT strain serving as a control to assess the development of cross-resistance in the resistant strain.

Table 1 Mode of action and bacterial resistance pathways of commonly used antibiotics.

Categorization	Antibiotics	Acronyms	Mode of action	Resistance pathways	
Penicillin	Ampicillin	AMP	Inhibits peptidoglycan biosynthesis	Hydrolysis, Efflux, Target changes	
Cephalosporins	Cefoperazone sodium	CFP	Inhibits peptidoglycan biosynthesis	Hydrolysis, Target changes, Pore protein deficiency	
Macrolides	Azithromycin	AZI	Inhibits protein synthesis	Hydrolysis, Glycosylation, Phosphorylation, Efflux, Target changes	
Aminoglycosides	Gentamycin	GM	Inhibits protein synthesis	Phosphorylation, Acetylation, Nucleotidylation, Efflux, Target changes	
Tetracyclines	Tetracycline sulfate	TET	Inhibits protein synthesis	Singleton-oxygenation, Efflux, Target changes	
Quinolones	Nalidixic acid, norfloxacin, enoxacin	PPA, NA, NFX, ENX	Inhibits DNA replication	Acetylation, Efflux, Target changes	
Carbapenems	Imipenem, Ertapenem, meropenem	IPM, ETP, MEM	Inhibits cell wall mucopeptide synthesis	Hydrolyze, Target changes, Pore protein deficiency	

Determination of growth curves

The test strains included WT E. coli K12 MG1655, as well as induced strains with minimum inhibitory concentrations of 1, 4, 8, and 16 µg/mL. The strains were incubated in MH medium containing subinhibitory concentrations of imipenem at 37 °C, with the optical density at 600 nm (D600 nm) measured every hour. Each strain was assessed in triplicate.

Swimming and swarming ability of wild and resistant strains

The swimming and swarming abilities were assessed following the method outlined by Liu et al. (2015), with slight modifications. For the swimming motility test, 0.3% agar and 0.5% glucose were added to the LB medium, while for the swarming motility test, 0.5% agar and 0.5% glucose were used. Single colonies of both the WT and drug-resistant strains were selected and cultured in LB broth until reaching a D600 nm of 1.0. Subsequently, 2 µL of bacterial solution was placed in the center of the respective culture media, followed by an incubation period at 37 °C for 24 h. The strains’ morphology was observed, and the motility diameter was measured with 3 replicates per strain to analyze differences in motility.

Determination of biofilm formation capacity by crystalline violet staining

In this study, crystal violet was utilized to stain biofilms to detect the biofilm formation capability of both wild strains and drug-resistant strains at the same D600 nm value (Coffey & Anderson, 2014). The procedure involved adding 198 µL of culture solution to each well of a 96-well polystyrene microplate, along with 2 µL of sample bacterial solution (0.5 McFarland turbidity), followed by incubation at 37 °C for 36 h. Subsequently, the culture solution was aspirated, sterile water (200 µL) was added to each well for a triple wash, and methanol (200 µL) was introduced to fix the biofilms for 15 min. After aspirating the methanol and air-drying the wells, a 1% crystal violet solution (200 µL) was added to each well and left to stain at room temperature for 5-10 min. Following the removal of excess dye and drying of the plate, 33% glacial acetic acid solution (200 µL) was added to each well and incubated at 37 °C for 30 min to dissolve the crystal violet. The absorbance value of the solution in the culture wells was measured using a microplate reader at 590 nm, with the experiment being conducted in triplicate.

Scanning electron microscope observation of bacterial morphology

A field emission scanning electron microscope (FESEM) was utilized to examine the morphological changes in E. coli caused by antibiotics. The study observed the WT strain, the imipenem-induced drug-resistant strain IPM-R. The procedure involved fixing the collected cells with 2.5% glutaraldehyde overnight at 4 °C, followed by dehydration with ethanol solutions of varying concentrations (30%, 50%, 70%, and 90%) for 10 min each, and finally 100% isoamyl acetate for 30 min before drying at 50 °C for 5 h. Subsequently, the cells were affixed to a FESEM holder and gold-plated via vacuum sputtering. The cell morphology was then observed using FESEM Verios 460.

Antibiotic action site mutations and carbapenem hydrolase gene detection in resistant strains

The mechanism of action of imipenem involves binding to penicillin-binding proteins, specifically PBP1-7. Spratt & Pardee (1975) conducted a study on the binding affinity of the parent compound thiomycin to E. coli penicillin-binding proteins (PBPs) and their impact on cell morphology. It has been established that PBPs in E. coli are crucial for cell replication and are involved in the biosynthesis of peptidoglycan sacculi. Another mechanism of bacterial resistance is the presence of a carbapenem hydrolase gene in the genome or plasmid of the strain, which confers antibiotic resistance (Paterson, 2006). The genomic DNA of drug-resistant strains was extracted and the penicillin-binding protein genes (PBP1-7) were amplified. The amplified genes and the corresponding primers are listed in Table S2. Additionally, carbapenem hydrolase genes were amplified for both drug-resistant and WT strains, with the amplified genes and primers listed in Table S3.

Transcriptome sample preparation

According to the growth curve, both the WT strain and the IPM-R strain reached a D600 nm of 0.5 during the logarithmic growth phase. Cells were harvested once both strains attained this D600 nm value. The WT strain was cultured in MH medium at 37 °C without antibiotics, while the IPM-R strain was cultured under the same conditions but with subinhibitory concentrations of imipenem. Following the collection of bacterial cells, they were sent to Sangon Biotech Co., Ltd. for RNA extraction and cDNA library construction. The results of the RNA extraction and cDNA library construction are presented in the attachment, and all procedures met the sequencing requirements.

Gene expression calculation and differential gene expression statistics

Sequencing samples are set in three biological replicates, and the statistical power of this experimental design, calculated in RNASeqPower is 0.84. To ensure comparability across different genes and experiments, researchers introduced the concept of Transcripts Per Million (TPM) (Jin, Wan & Liu, 2017). TPM quantifies the relative abundance of a specific transcript in the RNA pool, considering sequencing depth, gene length, and sample impact on read counts. It is a widely used method for estimating gene expression levels. The TPM calculation formula is as follows:

TPMi=XiLi∗1∑jXiLi∗106

Xi=total exon fragment/reads

Li=exon lengthKB.

The RPKM value for each gene in the three samples was calculated and used to determine the logarithm (fold change) as log2 (RPKM-resistant strain/RPKM parental strain). DEGseq software was employed to identify differentially expressed genes (DEGs) using a cutoff threshold of q Value ≤ 0.05 and |log2Fold Change | ≥ 1 (Wang et al., 2010).

Gene knockout strain preparation

Following the method described by Lau et al. (2023), the pdm4 suicide plasmid was utilized to knock out the target gene. This suicide plasmid necessitates the involvement of specific proteins for replication, rendering it unable to replicate in standard bacterial strains. Instead, it must be integrated into the bacterial chromosome and replicated alongside it. By leveraging this characteristic, the fragments flanking the gene intended for deletion are cloned into the suicide plasmid. After conjugation transfer, the homologous fragments on the suicide plasmid integrate with their counterparts in the bacterial genome. Through homologous recombination, precise deletion of the target gene can be accomplished.

Drug susceptibility assays

The MIC was determined following the method outlined by Feng et al. (2020), utilizing a fluorescence spectrophotometer to measure absorbance at D600 nm. The MIC was defined as the lowest drug concentration that completely inhibited bacterial growth in the well. Each test was conducted with three replicate wells, and the experiment was repeated at least three times. The experimental results were evaluated following the CLSI 2022 standard.

Statistical analysis

This article uses GraphPad Prism 9.0 to conduct statistical analysis of experimental data, and the t-test is used for specific analysis of data. * represents P < 0.05, which means the difference is significant; ** represents P < 0.01, which means the difference is very significant; *** represents P < 0.001, which means the difference is extremely significant; **** represents P < 0.0001, which means the difference is more significant than ***; ns means the difference is not significant.

Results

Induction and biological characterization of imipenem-resistant E. coli

Induction of imipenem-resistant E. coli

The WT strain K12 MG1655 was subjected to induction with sub-inhibitory concentration drugs, as illustrated in Fig. 1. Subsequently, a strain with a MIC of 16 µg/mL was produced (Fig. 2). Following the bacterial resistance assessment guidelines outlined in the CLSI 2022 version, a strain exhibiting an inhibition zone diameter greater than or equal to 23 mm is classified as drug-resistant and denoted as IPM-R (Table 2 and Fig. S1). The resistance level of the strain increased by over 500 times compared to its pre-induction state. The quality control strain used was E. coli ATCC 25922. However, it is worth noticing that the drug-resistant bacteria induced in this study did not have stable resistance. When cultured in a medium without imipenem, the MIC broke off at the third passage and returned to the same level as wild bacteria.

Figure 2 Changes in the minimum inhibitory concentration during the induction of imipenem-resistant strains.

Starting from the initial culture, the wild-type strain developed a drug-resistant variant with a minimum inhibitory concentration of 16 after 16 passages.

Table 2 Inhibition zone diameter of imipenem by different strains of bacteria.

Drug	Inhibition zone diameter (mm)	Inhibition zone diameter breakpoint (mm)	
	WT K-12	ATCC 25922	IPM-R	Sensitive	Resistance	
Imipenem	30.098	28.547	9.928	≥23	≤19	
29.678	28.583	10.853	
29.069	28.242	10.389	

Comparison of growth curves between wild and induced strains

This study utilized the 1/2 MIC induction method to generate strains with varying minimum inhibitory concentrations, followed by an analysis of their growth curves. The findings indicated that the growth rate of the induced strain was slower compared to the parent strain. Additionally, a correlation was observed between the drug concentration and the inhibitory effect on bacterial growth, as depicted in Fig. 3.

Figure 3 Growth curves of wild-type E. coli and strains with different minimum inhibitory concentrations.

According to the CLSI 2022 criteria for determining imipenem-resistant bacteria, the MIC breakpoints are set at 4 and 1 µg/mL, respectively. This study selected strains with minimum inhibitory concentrations of 1, 4, 8, and 16 µg/mL, as well as the wild type. The growth curves of these strains were evaluated while they were cultured in MH medium containing the corresponding subinhibitory concentrations. The results indicated that as the drug concentration increased, bacterial inhibition intensified, leading to a slower growth rate.

Cross-resistance in imipenem-resistant bacteria

Table 3 displays the drug susceptibility test outcomes for WT and drug-resistant strains. IPM-R strains exhibit cross-resistance to multiple antibiotics, such as β-lactams, lactones, tetracyclines, quinolones, and carbapenems, when compared to WT E. coli. Although antibacterial drugs may not belong to the same class, they can exhibit commonalities in their resistance mechanisms. Certain multidrug efflux systems and specialized barrier structures can facilitate the efflux of multiple antibiotics, resulting in bacterial cross-resistance. Transcriptomic analyses revealed that, in response to the selective pressure exerted by imipenem, E. coli increased the expression of seven transporters by more than 1.5 times. Notably, the ABC-type multidrug transport system AcrB has been confirmed as an active efflux pump contributing to bacterial multidrug resistance, which serves as a defense mechanism for bacteria (Tam et al., 2021). Furthermore, specific strains or communities possess unique barrier structures, such as biofilms and outer membrane proteins, which provide additional protection for bacteria. Structural and functional alterations in the cell wall and cell membrane can also lead to reduced permeability. Consequently, these non-specific resistance mechanisms are significant contributors to cross-resistance, as discussed in ‘Transcriptional Changes in Cell Movement and Communication’.

Table 3 Drug susceptibility testing of wild-type and drug-resistant strains.

Drugs	Minimum inhibitory concentrations (MIC) (µg/mL)	MIC breakpoint (µg/mL)	
	WT E. coli	IPM-R	ATCC 25922	Sensitive	Resistance	
Ampicillin	8	16	8	8	16	
Cefoperazone sodium	0.5	2	0.5	16	64	
Azithromycin	2	4	2	16	32	
Gentamycin	2	2	2	4	16	
Tetracycline	0.5	1	0.5	4	16	
Nalidixic acid	4	8	4	16	32	
Norfloxacin	0.06	0.25	0.06	4	16	
Enoxacin	0.25	0.25	0.125	2	8	
Imipenem	0.03	16	0.03	1	4	
Ertapenem	0.06	8	0.06	0.5	2	
Meropenem	0.03	0.5	0.03	1	4	

Swimming and swarming ability of wild and resistant strains

The swimming and swarming abilities of bacteria were assessed by measuring the lawn diameter after 24 h of culturing on semi-solid mediums. Figure 4 displays the culture results of both the strains before and after induction, while Fig. 5 presents the lawn diameter measurements. Comparing the induced drug-resistant bacteria with the parent strains, it was observed that the diameter of the bacterial lawn significantly decreased post-imipenem induction, indicating a reduction in bacterial motility. This is also consistent with the findings presented in ‘Observation of Strain Morphology by Field Emission Scanning Electron Microscopy’. Once bacteria form a biofilm, their mobility is significantly reduced (Tu et al., 2019).

Figure 4 Growth of wild and resistant strains on semi-solid medium.

Drug-resistant strains and wild-type strains exhibiting a minimum inhibitory concentration of 16 µg/mL were selected and subsequently tested for their motility on a plate devoid of antibiotics. The results indicated a significant reduction in the motility of the drug-resistant strains.

Figure 5 Lawn diameters of wild and drug-resistant strains.

Asterisks (****) indicate a p-value less than 0.0001.

Biofilm forming ability of wild and drug-resistant strains

When bacteria form biofilms, they secrete more extracellular matrix and aggregate. The staining intensity is typically deeper in biofilm-forming strains compared to those that do not form biofilms. Crystal violet staining was used to detect the biofilm formation of different strains over time. Results indicated that the biofilm formation ability of IPM-R was significantly enhanced compared to the wild strain at 24 h, and this ability increased over time (Fig. 6), suggesting that drug-resistant bacteria induced by imipenem form more biofilms to survive in adverse environments.

Figure 6 Comparison of biofilm-forming ability between wild and resistant strains.

The biofilm formation capability of the strain was assessed using crystal violet staining. The results indicated a significant enhancement in biofilm formation beginning at 24 h. Asterisks (****) indicate a p-value less than 0.0001.

Observation of strain morphology by field emission scanning electron microscopy

The scanning electron microscopy results depicted in Fig. 7 reveal the cell morphology. The WT strains show cells with a smooth surface, short rod shape, and uniform size (Figs. 7A and 7B). Upon drug induction (Figs. 7C and 7D), the cells exhibited significant swelling, presenting spherical or irregular shapes, some with damaged surfaces displaying wrinkles and defects. The shapes varied in size and irregularity, with many cells aggregating and an increase in the extracellular matrix. It was evident that a majority of the bacteria induced by imipenem were in a state of biofilm formation.

Figure 7 Scanning electron microscope observation of strain morphology.

Scanning electron microscopy revealed that, compared to the wild-type strain, the drug-resistant strain exhibited clustering and deformation. Furthermore, the results of the crystal violet staining indicated that the drug-resistant strain was present in the form of a biofilm.

Antibiotic action site mutations and carbapenem hydrolase gene detection in resistant strains

This study investigated the mutations in the PBP1-7 genes and the presence of carbapenem hydrolase genes, only the penicillin adapter protein PBP6b was found to be mutated (Figs. 8–9). PBP6b encodes a protein with carboxypeptidase activity, which is crucial for maintaining the shape of E. coli. Deletion of carboxypeptidase can result in morphological defects (Peters et al., 2016). The functional domain PBP5-C at amino acids 285-376 (i.e., 855–1,128 bp) of PBP6b is important for penicillin affinity. Amino acid substitutions in PBP5-C have been shown to impact β-lactam antibiotic resistance in vitro (Zhou et al., 2015). The sequencing results indicated that the 124th amino acid of PBP6b was mutated from glycine to valine. Additionally, the 350th amino acid was mutated from glutamic acid to aspartic acid, the 351st amino acid was mutated from isoleucine to serine, the 355th amino acid also underwent a mutation from isoleucine to serine, and the 364th amino acid was mutated from proline to serine, these mutations are hypothesized to contribute to resistance to imipenem.

Figure 8 Sequence of nucleic acid mutations in the PBP6b gene of drug-resistant strains.

In this study, the penicillin-binding proteins PBP1-7 from both wild-type and resistant strains were sequenced, revealing that only the PBP6b protein exhibited mutations.

Figure 9 Amino mutations in the PBP6b gene of drug-resistant strains.

The active site (63, 66, and 129 aa), the binding site (232 aa), and the functional domain (282-373 aa) of the PBP6b protein are illustrated in the figure. The findings indicate that, with the exception of the mutation at amino acid position 124, all other four positions reside within the PBP5-C functional domain.

Drug efflux gene knockout and imipenem susceptibility assays

Transcriptome data reveals an up-regulation of putative drug transporters, as shown in Table 4, We constructed gene-deleted strains of E. coli for each of these genes, the electrophoretic patterns of agarose gel for the target gene deletion strains are illustrated in Figs. S2–S5, and the primers used for PCR are shown in Table S6. Sequencing results confirmed that all constructs were successful (Tables S4–S5). The acrB, ybhR, and ybhS gene deletion strains were derived from Feng et al. (2020). Subsequently, we assessed the changes in drug sensitivity of each strain to imipenem (Table 5). The results indicated that the deletion of various drug efflux genes, including acrB, did not significantly affect the minimum inhibitory concentration, suggesting that imipenem resistance is not primarily attributed to drug efflux mechanisms in sexual formation.

Table 4 Genes of upregulated putative drug transporter proteins.

Gene	ID	Fold change	q-value	Gene description	
mdtC	b2076	1.697016214	0.000125618	Multidrug efflux pump RND permease subunit MdtC	
mdtD	b2077	2.263291823	0.001064678	Putative multidrug efflux pump MdtD	
macB	b0879	1.540842543	7.93E−08	ABC-type tripartite efflux pump ATP binding/membrane subunit	
acrB	b0462	1.435441925	3.41E−07	Multidrug efflux pump RND permease AcrB	
mdtF	b3514	1.789857922	2.09E−05	Multidrug efflux pump RND permease MdtF	
ybhR	b0792	1.672548492	9.88E−05	ABC exporter membrane subunit YbhR	
ybhS	b0793	1.632804804	2.20E−05	ABC exporter membrane subunit YbhS	

Table 5 Changes in the minimum inhibitory concentration of imipenem by drug efflux gene deletion strains.

Strains	Minimum inhibitory concentrations (MIC) (µg/mL)	
K-12	0.125	
ATCC 25922	0.125	
ΔmdtC	0.125	
ΔmdtD	0.125	
ΔmacB	0.125	
ΔacrB	0.125	
ΔybhR	0.125	
ΔybhS	0.125	
ΔmdtF	0.060	

Comparative transcriptomic analysis of wild and drug-resistant strains

Statistics of differentially expressed genes

Using q value ≤ 0.05 and |log2Fold Change | ≥ 1 as screening criteria, a total of 1,260 differentially expressed genes were identified. Among them, 880 genes showed up-regulated expression in drug-resistant strain B compared to the original strain A, while 380 genes displayed down-regulated expression (Fig. 10).

Figure 10 Volcano plot statistics of differentially expressed genes in wild strain (A) and resistant strain (B).

In the differential expression volcano plot, each point represents a gene, with the x-axis indicating the log-arithm of the fold difference in gene expression between the two samples. A larger absolute value signifies a greater difference in expression folds between the samples. The y-axis represents the negative logarith-mic value of the false discovery rate, where a higher value indicates more significant differential expres-sion and greater reliability of the identified differentially expressed genes following screening. Tran-scriptomic data are provided in Table S1.

GO, COG, and KEGG classification of differentially expressed genes

To elucidate the functions of differentially expressed genes (DEGs), all unigenes were aligned with entries in the Gene Ontology (GO) database (Fig. 11). The classification outcomes indicate that drug-resistant bacteria exhibit significant differences from WT strains in terms of cell structure, substance metabolism, and transport (Figs. S6–S9). Figure 12 displays the top 20 most significantly enriched COG categories, with a focus on amino acid transport and metabolism, defense mechanisms, inorganic ion transport and metabolism, energy production, and conversion, translation, and ribosome structure and biosynthesis (Figs. S10–S11). Figure 13 displays only the most significantly enriched genes. The analysis of the top 30 KEGG pathways revealed a high concentration of genes in the ABC transport system (ko02010), amino acid biosynthesis (ko01230), and ribosome (ko03010) pathways, which were found to be significantly enriched (Fig. S12).

Figure 11 GO function classification.

The horizontal axis is the functional classification, and the vertical axis is the number of genes within that classification (right) and their percentage of the total number of genes on the annotation (left). Different colors represent different classifications. Light colors represent differential genes and dark colors represent all sequenced genes.

Figure 12 COG enrichment.

The vertical axis represents the functional annotation information, the horizontal axis represents the Rich factor corresponding to the function, the size of Qvalue is represented by the color of the dot, the smaller the Qvalue is, the color is closer to red, and the number of differential genes contained under each function is represented by the size of the dot.

Figure 13 KEGG pathway enrichment.

Transcriptional changes to nutrient uptake and metabolism

Under antibiotic stress, distinct transcriptional responses were observed in the nutrient assimilation pathways for carbon, sulfur, and nitrogen sources. In carbon metabolism, both conventional and unconventional carbohydrates (galactose, fructose, mannose, starch, and sucrose) are upregulated to varying extents (Figs. S13–S14). These findings collectively support the upregulation of metabolic pathways. In sulfur metabolism, the assimilation of sulfate reduced to hydrogen sulfide is upregulated (Fig. S15), in line with the upregulation of genes encoding ABC sulfate and thiosulfate transporters (Fig. S34), indicating global assimilation of sulfur that was upregulated. Furthermore, the sulfur transfer system was also upregulated (Fig. S16). Nitrogen metabolism, including nitrate and nitrite, exhibited an upregulation (Fig. S17).

The results align with changes in gene expression of the KEGG pathway ABC transporter (Eco02010) and phosphotransferase system (Pts) transporter (Eco02060), along with genes associated with transporter activity gene ontology term (GO 0005215). Overall, it is suggested that upon induction, there is an increase in the intake of nitrogen, sulfur, and phosphorus sources, a decrease in sugar intake, and an increase in the intake of amino acids and other carbon sources to supplement carbon intake. The degree of upregulation of efflux proteins following the ingestion of peptides (>5.0) was significantly higher than that observed for other nutrients (1.0–2.0).

Transcriptomic changes in energy-producing metabolic pathways

Polysaccharides, particularly glycogen, can function as energy reserves for E. coli. In response to antibiotic stress, the metabolic pathways of glycogen are transcriptionally upregulated, similar to other carbohydrate pathways (Fig. S18). Furthermore, the PTS system, a multi-protein phosphorylation system located in the cell membrane, integrates carbohydrate transport and phosphorylation. Genes encoding transporters for β-glucoside, N-acetylmuramic acid, galactose, and fructose were found to be up-regulated in the PTS system, while no down-regulation of genes was observed (Fig. S21).

In the glycolytic pathway from glucose to pyruvate, the enzymes ptkA (6-phosphofructokinase) and pykF (pyruvate kinase) were down-regulated about 1.0 to 1.5 times, both being rate-limiting enzymes. Subsequently, the TCA cycle showed a general up-regulation (Fig. S19). Within the pentose phosphate pathway, the key genes zwf, pgl, and gnd in the first stage from glucose-6 phosphate to ribose-5 phosphate remained unchanged. In the second stage, the genes encoding transaldolase (talA, talB) and transketolase were unchanged, while tktA and tktB were up-regulated (Fig. S20). Beta-oxidation and the acyl-CoA cycle were significantly upregulated more than 2.0 times during fatty acid degradation (Fig. S22), leading to the production of a large amount of acetyl-CoA for entry into the tricarboxylic acid cycle, generating substantial energy. Both sugars and non-sugar carbohydrates ultimately enter the TCA cycle for energy production. The TCA cycle exhibited significant up-regulation (2.0–4.0) after antibiotic induction (Fig. S23), suggesting that post-induction, the bacteria generated more energy, possibly shifting from glucose-derived energy to utilizing other carbon sources for energy production.

Transcriptional changes in biosynthesis

Under antibiotic stress, significant changes have been observed in the biosynthetic pathway of amino acids, leading to an up-regulation of amino acid biosynthesis (Figs. S24–S29). Among the 21 common amino acids, 13 species exhibit clear metabolic/biosynthetic regulation patterns. Notably, the biosynthetic pathways of histidine, tryptophan, valine, leucine, and isoleucine were significantly up-regulated. Additionally, alanine, aspartic acid, glutamic acid, arginine, proline, glycine, serine, threonine, histidine, cysteine, and methionine showed varying degrees of up-regulation. This upregulation aligns with the increased expression of genes encoding amino acid transporters in the ABC system (Fig. S34).

In protein biosynthesis, ribosome-encoding genes are typically down-regulated about 1.5 times (Fig. S30). Imipenem treatment leads to the down-regulation of nucleoside triphosphate (NTP) and their precursors biosynthesis, while deoxynucleoside triphosphates (dNTPs) show no significant effect. Notably, the dnaE gene encoding DNA polymerase is up-regulated 1.37 times, whereas rpoA encoding the α subunit of RNA polymerase is down-regulated 1.1 times (Fig. S31). This suggests that drug-resistant bacteria down-regulate intracellular transcription reactions and protein synthesis processes while increasing amino acid synthesis to meet energy needs or cell wall biosynthesis and other growth essentials.

Transcriptional changes in cell membrane components

In the context of cell wall biosynthesis, lipopolysaccharide synthesis remained relatively stable (Fig. S32). Following exposure to the cell wall inhibitor imipenem, there was a notable increase in DAP-type epidermal polysaccharide synthesis within peptidoglycan synthesis, with an overall significant up-regulation observed. Imipenem specifically targets the penicillin-binding protein PBP2 which is upregulated 1.12 times (Fig. S33). The peptidoglycan molecule serves as a key constituent of the bacterial cell wall, characterized by a network of N-acetylglucosamine and N-acetylmuramic acid linked by β-1,4 glycosidic bonds and short peptide tails. The presence of E. coli PBP2 plays a crucial role in maintaining the rod-shaped structure of the bacteria, as its reduction or absence can lead to spherical transformation, ultimately resulting in bacterial dissolution and death. Consequently, it is postulated that there is an elevation in the synthesis of essential cell wall materials required for bacterial survival.

Transcriptional changes in membrane transporter proteins

Genes within the KEGG pathway ABC transporter (Eco02010) and phosphotransferase system (Pts) transporter (Eco02060), along with genes associated with transporter activity gene ontology term (GO 0005215), were examined.

In the ABC transport system (Fig. S34), the uptake of sugars such as maltose, sorbitol, mannitol, ribose, and xylose are down-regulated (1.0–1.5). Conversely, the uptake of other carbon sources like L-arabinose, methyl galactopyranoside, and glycerophosphate is upregulated (>2.0). Additionally, the uptake of sulfur sources such as sulfates, sulfonates, and thiamine is upregulated (>2.0), while the uptake of nitrogen sources like putrescine and phosphorus sources like phosphate is up-regulated (>2.0). Moreover, the uptake of amino acids such as arginine, glutamic acid, aspartic acid, cysteine, methionine, and branched-chain amino acids is up-regulated (1.0–1.5), and the intake of peptides like oligopeptides, secondborn, cationic peptides, and glutathione is upregulated (>2.0). Furthermore, the intake of trace elements like nickel ions is increased (>2.0), while zinc ions are decreased by 1.56 times. In the phosphatase transfer system (Fig. S35), the transfer of β-glucoside, N-acetylmuramic acid, galactose, fructose, and nitrogen is up-regulated less than 2.0 times, with β-glucoside and N-acetylmuramic acid being important substances for synthetic peptide polypeptides. Additionally, the cell membrane contains numerous transporters that aid in removing harmful substances from the cell.

Transcriptional changes in cell movement and communication

Cell motility, communication, flagellar assembly, biofilm formation, quorum sensing, and bacterial chemotaxis are common stress responses in bacteria. Flagellum assembly remained unchanged (Fig. S36). In the quorum sensing pathway, the upregulation of luxS leads to an upregulation of the AI-2 transport system (Figs. S37–S38). Among them, the luxS gene was upregulated 1.23 times, while the AI-2 transport system exhibited an upregulation of more than 2.0 times. During bacterial chemotaxis (Fig. S39), Tsr and Trg, which are methyl-accepting chemotactic proteins, were up-regulated about 1.0 times. Along with the substrate-binding proteins MglB (for the methylgalactoside transport system) and DppA (for the dipeptide transport system), were up-regulated 3.0 times.

Discussion

Bie et al.’s (2023) study categorized E. coli’s resistance mechanisms to various antibiotics into four distinct types: pessimists, defenders, evaders, and ignorance. Pessimists involve E. coli ceasing unnecessary activities, activating efflux pumps, and enhancing biofilm formation, resembling a hibernation state. Defenders exhibit an upregulation of protein and nucleotide biosynthesis to counter threats. When faced with cell wall inhibitors, E. coli intensifies cell wall biosynthesis, while non-cell wall inhibitors prompt increased mobility and biofilm formation, characteristic of evaders. The ignorant category refers to E. coli maintaining a stable transcriptome despite antibiotic exposure.

Inspired by the research of Bie et al. (2023), it is suggested that E. coli’s resistance strategy to imipenem falls under the defender category. When faced with imipenem, the downregulation of ribosomal genes in drug-resistant bacteria leads to decreased intracellular protein synthesis. Imipenem disrupts the penicillin-binding site and inhibits peptidoglycan network formation, prompting drug-resistant bacteria to upregulate the peptidoglycan pathway. In quorum sensing, the luxS gene of the LuxS/AI-2 system is upregulated, facilitating biofilm formation. Energy production pathways show upregulation of amino acid and unconventional carbohydrate metabolism in drug-resistant bacteria, while the metabolism of conventional carbohydrates like glucose is downregulated, possibly due to the biofilm state.

When exposed to imipenem, E. coli decreases the synthesis of intracellular proteins, enhances the production of damaged peptidoglycan, promotes biofilm formation, upregulates the expression of efflux pumps, and enters a dormant state. Biofilm formation and mutations of the PBP6b gene are identified as the primary mechanisms bacteria employ to resist imipenem, offering a foundation for exploring new drug targets and inhibitor development. The transcriptomic analysis conducted in this study elucidated the alterations in gene expression induced by imipenem stress in E. coli. Nevertheless, additional experiments, including whole genome sequencing and protein phosphorylation profiling, are essential to uncover the resistance mechanisms associated with imipenem.

Supplemental Information

Figure S1 Imipenem inhibition zone test

Figure S2 PCR validation of mdtC knockout strains

Column 15 is the wild-type strain control, and the remaining columns are knockout strain verifications. The strains used in sequencing and experiments are in column 3 (the 5000 bp marker in the left column is not considered). The primers used were mdtC-2 listed in Table S6.

Figure S3 PCR validation of mdtD knockout strains

The 1st column is the verification of mdtD in the wild-type strain, the 2–3 columns are the verification of the mdtD gene deletion strain; the 5th column is the verification of the mdtC-1 primer in the wild-type strain, and the 6–8 columns are mdtC-1 Verification of primers in gene deletion strains; column 9 is the verification of mdtC-2 primers in wild-type strains, columns 10–12 are verification of mdtC-2 primers in gene deletion strains (the 5000 bp marker in the left column is not considered).

Figure S4 PCR validation of macB knockout strains

The 1st column is the verification of mdtC-1 in the wild-type strain, the 2-3 columns are the verification of the mdtC-1 gene deletion strain; the 5th column is the verification of the mdtC-2 in the wild-type strain, and the 6-8 columns are verification of mdtC-2 in gene deletion strains; column 9 is the verification of macB primers in wild-type strains, columns 10-16 are verification of mdtC-2 primers in gene deletion strains(the 5000 bp markers in the figure are not considered).

Figure S5 PCR validation of mdtE and mdtF knockout strains

The 1st column is the verification of mdtE in the knockout strain, the 2nd column is the verification of mdtE in the wild-type strain; the 3rd column is the verification of mdtF in the knockout strain, and the 4th column is the verification of mdtF in the wild-type strain.

Supplemental Information 1 Supplemental figures

Table S1 Differently expressed genes

Table S2 Penicillin-binding protein gene and primer sequences

Table S3 Carbapenem hydrolase gene and primer sequences

Table S4 Sequencing of the up-regulated efflux protein gene deletion strain

Table S5 Sequencing of the up-regulated efflux protein gene wild-type strain

Table S6 Primers for agarose gel electrophoresis of gene knockout strains

Supplemental Information 13 Growth curves of wild-type E. coli and strains with different minimum inhibitory concentrations

Supplemental Information 14 Growth of wild and resistant strains on semi-solid medium

Supplemental Information 15 Lawn diameters of wild and drug-resistant strains

Supplemental Information 16 Comparison of biofilm-forming ability between wild and resistant strains

We express our gratitude to all those who have contributed to the research, as well as to the reviewers who participated in the publication process.

Additional Information and Declarations

Competing Interests

Author Contributions

Data Availability

The authors declare there are no competing interests.

Chunyu Tong conceived and designed the experiments, prepared figures and/or tables, and approved the final draft.

Yimin Liang conceived and designed the experiments, performed the experiments, analyzed the data, prepared figures and/or tables, and approved the final draft.

Qi Liu performed the experiments, prepared figures and/or tables, and approved the final draft.

Honghao Yu analyzed the data, prepared figures and/or tables, and approved the final draft.

Wenzhi Feng analyzed the data, prepared figures and/or tables, and approved the final draft.

Bocui Song conceived and designed the experiments, authored or reviewed drafts of the article, and approved the final draft.

The following information was supplied regarding data availability:

The raw sequences are available at the Genome Sequence Archive in National Genomics Data Center (Chen et al., 2021; Yongbiao et al., 2022), China National Center for Bioinformation/Beijing Institute of Genomics, Chinese Academy of Sciences: PRJCA026645.

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
