# Peer review of "Resistance characterization and transcriptomic analysis of imipenem-induced drug resistance in Escherichia coli"

_PeerJ, doi:10.7717/peerj.18572_

## Round 0.1 · original submission · Major Revisions

The authors are requested to carefully revise the manuscript and answer the questions raised by the reviewers.

·

Basic reporting

- Figure legends are not verose enough and need to be expanded to provide essential information.
- check formatting for italics
- Results, Discussion and Conclusions contain a number of redundancies.

Experimental design

The authors carry out a series of basic phenotype experiments to elucidate changes in response to antibiotic resistance. Subsequently, a transcriptomicanalysis is carried out to investigate changes in gene expression.
- please provide actual data (i.e., calculate the growth rates) for the strains tested for growth. Further, it seems that all strains reach the same/similar OD after 12 hours.
- Table 2 shows changes in antibiotic resistance but many changes still remain below the MIC breakpoint. This findign should be discussed in more detail, especially in the light of implied cross-resistance.
- In Fig 3, it is unclear whether the strains were grown in the presence or absence of an inhibitor.
- line 269: what are 'positive DNA fragments'?
- Figure 8 shoudl be updated to include aa changes in relevant functional domains and active sites.
- no detailed information is provided on the exact experimental makeup for the (differential) transcriptomic analysis.
- Why are there 2 different coloured bars in Figure 10?
- line 379: what was the actual fold change in gene expression for pbp2?
- it is unclear why only a single gene (SBP2) was tested for nt changes. While the gene is a primary target, the random nature of non-targeted antibiotic resistance may have led to other changes across the genome in both coding and non-coding regions. WGS is required to elucidate whether observed changes in gene expression are also linked to other non-silent mutations.
- Are the observed transciptomic changes static or dynamic after addition of the antibiotic?
- There is no information provided on the genomic and transcriptomic changes of previous generations. Do the observed changes in gene expression appear all at once or do they gradually accumulate?
- Overall, the disucssion of the transcriptomic data and the correlation/mechanistic insights of the oberserved phenotypes remain superficial. For example, little detailed information is provided on relevant genes and operons involved in biofilm formation. Biofilms comprise a complex matrix of EPS, eDNA, and proteins.

Validity of the findings

Development and evolution of antibiotic resistance in E. coli has been investigated for several decades and the authors have missed an opportunity to interrogate results from both omics technologies and phenotype assays to bring together a more detailed and mechanistic picture.
As the submission is presented, little new information is provided and the authors would need to carry out additional experiments and a more in depth mechanisitc analysis of all datasets.

Reviewer 2 ·

Basic reporting

Tong and colleagues have obtained imipenem-resistant strains through laboratory evolution experiments and have sought to elucidate the underlying resistance mechanisms primarily through transcriptome analysis. Additionally, they have reported phenotypic changes in the isolated resistant strains, including alterations in biofilm formation, motility, and morphology. As noted in the Introduction, the emergence of imipenem-resistant strains is a serious issue, making the elucidation of resistance mechanisms, as pursued in this study, an important task. The transcriptome analysis revealed global changes in gene expression associated with resistance, and the authors have conducted GO classification, COG, and KEGG enrichment analyses, which are commendable. However, there are significant shortcomings in the presentation of results and discussion. In particular, the description of the transcriptome data often reads as a mere listing of gene expression changes, making it difficult to discern what is truly important for imipenem resistance. Beyond well-known phenomena such as the upregulation of efflux pumps and biofilm formation mentioned in lines 432-434 of the Discussion, it remains unclear what specific factors are critical to the development of imipenem resistance (and even the efflux pump findings are not well-described in the Results section). Moreover, the manuscript contains numerous typographical errors, and I strongly recommend a more thorough review of the text before resubmission.

Experimental design

(1) The transcriptome analysis section lacks clarity in both the Methods and Results sections regarding the specific strains, culture conditions, and growth phases used for total RNA extraction. Additionally, if the impact of antibiotic exposure was assessed (as suggested by the phrase "after antibiotic induction"), the timing, concentration, and conditions of antibiotic addition should be clearly stated to facilitate interpretation of the results.

(2) The authors have identified phenotypic changes, such as altered motility and biofilm formation, in addition to imipenem resistance. Given these findings, it would be valuable to conduct further analyses that link these phenotypic changes to the transcriptomic data.

Validity of the findings

(3) The transcriptional changes are described in a format like "upregulated (Supplemental Figure)," but it is difficult to gauge the extent of these changes without referring to the supplemental material. The main text should provide sufficient context to understand the key findings.

(4) Lines 333-334: Wild-type E. coli lacks the ability to utilize starch (Rosales-Colunga & Martinez-Antonio 2014 Microb Cell Fact 13:74), and since glucose is not added to the MH medium, it seems unlikely that glycogen synthesis would be active under these conditions. The impact of such metabolic changes on imipenem resistance should be discussed in more depth.

(5) Lines 334-336: If you claim that arbutin, salicylic acid, and ethanol can be used for growth, evidence should be provided, such as citations of reports demonstrating growth in minimal media or other relevant experimental results.

Additional comments

no comment

·

Basic reporting

This study investigates how wild-type Escherichia coli (E. coli) adapts to antibiotic stress, particularly under the selective pressure of imipenem, a broad-spectrum beta-lactam antibiotic. By subjecting the bacteria to continuous in vitro induction, the researchers were able to select strains that developed resistance to imipenem. These resistant strains also exhibited an enhanced ability to form biofilms, a common defense mechanism that helps bacteria survive in hostile environments, including those with antibiotics.

Experimental design

Methodology is described in details with well defined research question.

Validity of the findings

The study's results highlight a multifaceted strategy that E. coli employs to survive antibiotic stress. This includes metabolic adjustments that optimize resource use and enhance defense mechanisms. These insights could be crucial for understanding bacterial drug resistance and identifying new targets for antibiotic development, potentially leading to more effective treatments against resistant bacterial strains.

Additional comments

To delve deeper into the molecular mechanisms behind this resistance, the study employed transcriptomic sequencing. This technique allowed the researchers to observe the changes in gene expression within the resistant bacteria. The findings showed significant alterations in various metabolic pathways and cellular processes. Specifically, the bacteria reduced the synthesis and metabolism of non-essential substances, likely to conserve energy and resources, while simultaneously boosting activities that help resist external threats, such as nutrient assimilation, substance transport, and cell wall biosynthesis.

The manuscript is too lengthy to understand it at a time. I want some corrections with reference to english language, grammatical errors are present throughout the manuscript. Manuscript needs to be concise, short and crisp.
Repetitions are there in discussion and introduction section.

---

## Round 0.2 · Minor Revisions

The authors are requested to carefully revise the manuscript and answer the remaining questions raised by the reviewers.

·

Basic reporting

The authors have provided an overall improve revised manuscript

Experimental design

The newly presented gene knockouts are appreciated and provided additional depth to the manuscript.
- the successful gene knockout still needs to be verified by PCR
- one of the major comments, the absence of WGS to determine genetic changes of the evolved strains is still missing. The gene knockout data are not a substitute to investigate the underlying mechanisms.

Validity of the findings

Overall, the presented data is valid with some gaps remaining.

Additional comments

WGS should be carried out and included in the data.

Reviewer 2 ·

Basic reporting

The authors have adequately addressed my comments.
E. coli should be italicized.

Experimental design

The authors have adequately addressed my comments.

Validity of the findings

The authors have adequately addressed my comments.

·

Basic reporting

The overall manuscript has been revised and condensed in order to better understand this study. Authors have performed some new experiments like they performed drug susceptibility testing by knockout of the up-regulated efflux gene to check the effect of drug efflux on resistance.

Experimental design

The results have been rewritten to some extent and looks more clear and accurate now.

Validity of the findings

The authors has performed some experiments for drug susceptibility testing by knockout of the up-regulated efflux gene and found that the effect of efflux on imipenem resistance was very small, although it will add substantial value to the findings.

Additional comments

NA

---

## Round 0.3 · Minor Revisions

The authors are requested to carefully revise the manuscript and answer the issue raised by the reviewer.

·

Basic reporting

Thank you for adding the additional experimental validation of the gene knockouts.

However, a new comment by the authors raised some concern: "The drug-resistant bacteria we induced cannot be stored for a long time and need to be induced again. "
If these adapted strains show such remarkable genetic instability that they cannot be stored without loss or significant changes to the respective genotypes, then this must be reported in detail.

Experimental design

OK, with above comments under consideration

Validity of the findings

OK, with above comments under consideration

Additional comments

none

---

## Round 0.4 · accepted · Accept

After revisions, two reviewers agreed to publish the manuscript. There is one reviewer left with a minor revision, and I think the author has responded adequately. I also reviewed the manuscript and found no obvious risks to publication. Therefore, I also approved the publication of this manuscript.